# Epidemiology and Outcome of Early-Onset Acute Kidney Injury and Recovery in Critically Ill COVID-19 Patients: A Retrospective Analysis

**DOI:** 10.3390/biomedicines11041001

**Published:** 2023-03-23

**Authors:** Alice Ruault, Carole Philipponnet, Vincent Sapin, Bertrand Evrard, Radhia Bouzgarrou, Laure Calvet, François Thouy, Kévin Grapin, Benjamin Bonnet, Mireille Adda, Bertrand Souweine, Claire Dupuis

**Affiliations:** 1CHU Clermont-Ferrand, Service de Réanimation Médicale, F-63000 Clermont-Ferrand, France; 2CHU Clermont-Ferrand, Service de Néphrologie, F-63000 Clermont-Ferrand, France; 3CHU Clermont-Ferrand, Laboratoire de Biochimie, F-63000 Clermont-Ferrand, France; 4CHU Clermont-Ferrand, Laboratoire d’Immunologie, F-63000 Clermont-Ferrand, France; 5ECREIN, UMR1019 UNH, UFR Médecine de Clermont-Ferrand, Université Clermont Auvergne, F-63000 Clermont-Ferrand, France; 6Laboratoire Microorganismes: Génome et Environnement, UMR CNRS 6023, Université Clermont Auvergne, F-63000 Clermont-Ferrand, France; 7Unité de Nutrition Humaine, INRAe, CRNH Auvergne, Université Clermont Auvergne, F-63000 Clermont-Ferrand, France

**Keywords:** acute kidney injury, COVID-19, outcome, intensive care

## Abstract

Background: The clinical significance of early-onset acute kidney injury (EO-AKI) and recovery in severe COVID-19 intensive care unit (ICU) patients is poorly documented. Objective: The aim of the study was to assess the epidemiology and outcome of EO-AKI and recovery in ICU patients admitted for SARS-CoV-2 pneumonia. Design: This was a retrospective single-centre study. Setting: The study was carried out at the medical ICU of the university hospital of Clermont-Ferrand, France. Patients: All consecutive adult patients aged ≥18 years admitted between 20 March 2020 and 31 August 2021 for SARS-CoV-2 pneumonia were enrolled. Patients with chronic kidney disease, referred from another ICU, and with an ICU length of stay (LOS) ≤72 h were excluded. Interventions: EO-AKI was defined on the basis of serum creatinine levels according to the Kidney Disease Improving Global Outcomes criteria, developing ≤7 days. Depending on renal recovery, defined by the normalization of serum creatinine levels, EO-AKI was transient (recovery within 48 h), persistent (recovery between 3 and 7 days) or AKD (no recovery within 7 days after EO-AKI onset). Measurements: Uni- and multivariate analyses were performed to determine factors associated with EO-AKI and EO-AKI recovery. Main Results: EO-AKI occurred in 84/266 (31.5%) study patients, of whom 42 (50%), 17 (20.2%) and 25 (29.7%) had EO-AKI stages 1, 2 and 3, respectively. EO-AKI was classified as transient, persistent and AKD in 40 (47.6%), 15 (17.8%) and 29 (34.6%) patients, respectively. The 90-day mortality was 87/244 (35.6%) and increased with EO-AKI occurrence and severity: no EO-AKI, 38/168 (22.6%); EO-AKI stage 1, 22/39 (56.4%); stage 2, 9/15 (60%); and stage 3, 18/22 (81.8%) (*p* < 0.01). The 90-day mortality in patients with transient or persistent AKI and AKD was 20/36 (55.6%), 8/14 (57.1%) and 21/26 (80.8%), respectively (*p* < 0.01). MAKE-90 occurred in 42.6% of all patients. Conclusions: In ICU patients admitted for SARS-CoV-2 pneumonia, the development of EO-AKI and time to recovery beyond day 7 of onset were associated with poor outcome.

## 1. Introduction

Since the beginning of the outbreak of the severe acute respiratory syndrome coronavirus 2 (SARS-CoV-2), many studies have been published to better understand and characterize the global pandemic and its impact on the healthcare system especially in intensive care units (ICUs).

We now know that the virus does not only affect the respiratory system but also other organ systems, with acute kidney injury (AKI) probably being the second most frequent organ failure [1,2,3].

Although early reports from China showed a low AKI rate amongst patients [4,5,6] with SARS-CoV-2 pneumonia, recent studies suggest a higher average incidence, between 44% and 85% in critically ill patients, and a high mortality rate rising up to 51% [7,8,9,10,11,12,13,14,15] in cases of AKI (Appendix A).

Several risk factors of AKI in Coronavirus Disease-19 (COVID-19) have been reported such as male gender, ethnicity, older age and pre-existing long-term conditions including diabetes, obesity and chronic kidney disease [1,7,16,17,18]. Several mechanisms could explain the occurrence of AKI in COVID-19 including acute systemic inflammation, coagulopathy, direct viral toxicity, renin angiotensin system activation or unspecific kidney injury factors, such as lung/heart dysfunction, hypovolemia, a high level of positive end-expiratory pressure (PEEP), hemodynamic instability or nephrotoxic drugs [19,20,21,22].

Studying renal recovery to characterize AKI has been suggested in the last few years [23,24,25] as a means to predict patient outcomes and thereby improve treatment and prognosis. For instance, Abdel-Nabey et al. [26] underlined the importance of assessing AKI and renal recovery in order to target a specific population at a higher risk of residual kidney dysfunction.

Until recently, very few studies focused on renal recovery including consequences and risk factors in patients with SARS-CoV-2 pneumonia [16,27].

Thus, the purpose of the study, called SarSCoV-AKI, was to assess the prevalence, risk factors and outcome of early-onset AKI (EO-AKI) and recovery in critically ill patients admitted for severe COVID-19.

## 2. Materials and Methods

### 2.1. Study Design

The SarSCoV-AKI study was a retrospective analysis of prospectively collected data. It was a single-centre observational study performed at the medical ICU of the university hospital of Clermont-Ferrand, France. All consecutive adult patients aged ≥18 years admitted between 20 March 2020 and 31 August 2021 with a positive SARS-CoV-2 polymerase chain reaction test result, pneumonia confirmed by computed tomography and hypoxemia requiring >6 L/min supplemental oxygen were enrolled. Exclusion criteria were patients referred from another ICU, patients with chronic kidney disease stages 4 and 5, kidney transplant recipients, patients who refused to participate and patients with a hospital stay shorter than 72 h since they could not be classified according to our definitions. If there were multiple admissions, only the first admission to the ICU was included.

### 2.2. Ethics

The SarSCoV-AKI study received approval from the ethics committee of the French Intensive Care Society in 2020 (CE-SRLF20-21—French acronym for Comité éthique de la Société de réanimation de langue française) in accordance with our local regulations. All patients or close relatives were informed that their data would be included in the SarSCoV-AKI cohort study and gave written informed consent for the storage and research use of residual blood from samples collected as part of routine care. All the procedures were followed in accordance with the ethical standards of the responsible committee on human experimentation (institutional or regional) and with the Helsinki Declaration of 1975.

### 2.3. Data Collection

All data were obtained from medical records and electronic patient charts. Baseline patient characteristics were collected, including demographics and comorbidities, before ICU admission. The variables recorded regarding ICU admission and treatments were relative to clinical presentation, reason for ICU admission, diagnosis, therapies implemented and outcomes. Blood sampling and routine biological testing were performed on the day of admission and daily according to standard laboratory protocols.

### 2.4. Definitions

AKI was defined according to the classification of Kidney Disease Improving Global Outcomes (KDIGO) [28] (Annex 1), using only the serum creatinine (SCr) component. “Baseline SCr” was the best outpatient SCr value between 7 and 365 days before ICU admission or, if unavailable, was estimated using the Modification of Diet in Renal Disease (MDRD) equation [28]. Early-onset AKI was defined as AKI occurring within 7 days after ICU admission: only the first AKI episode was taken into account in the study. Patients were stratified according to the highest AKI stage attained during this first episode.

EO-AKI recovery was based on a sustained (≥48 h) and complete reversal of AKI by KDIGO criteria, and therefore a minimum of 48 h of renal recovery was necessary to separate two distinct AKI episodes. EO-AKI was classified as transient, persistent and AKD according to the Acute Dialysis Quality Initiative (ADQI) [25]. “Transient” AKI was defined as renal recovery within 48 h of AKI onset, and “persistent” AKI as renal recovery occurring ≥3 days and <7 days of EO-AKI onset. Acute kidney disease (AKD) was characterized when AKI stage 1 or greater persisted ≥7 days after EO-AKI onset [25] (Annex 2). Chronic kidney disease (CKD) staging was defined according to the KDIGO definition [28].

The outcome of major adverse kidney event at 90 days (MAKE-90) was the composite outcome of death, dialysis dependence or a glomerular filtration rare (eGFR) <60 mL/min 90 days after ICU admission [25,29,30]. GFR was estimated using Chronic Kidney Disease Epidemiology Collaboration (CKD-EPI) equation (31). SCr at 90 days was retrieved from the medical health records and if unavailable, the patient’s general practitioner was contacted.

Pulmonary bacterial co-infection was defined by the presence of a community-acquired or hospital-acquired bacterial pneumonia associated with SARS-CoV2 pneumonia on ICU admission. The presence of pulmonary bacterial co-infection was defined by the presence of radiological and/or scanographic condensation, bacteriological documentation (a positive quantitative culture of lower respiratory tract samples collected as recommended: bronchoalveolar lavage, >10^4^ CFU/mL, plugged telescoping catheter, >10^3^ CFU/mL, endotracheal aspirate, >10^6^ CFU/mL) and/or presence of positive antigenuria, as defined by the European Centre for Disease Control and Prevention [31]. If bacterial pneumonia occurred at least 2 days after intubation, it was classified as ventilator associated pneumoniae (VAP) [32]. The period at risk for VAP begins from 48 h after intubation, until removal of the tracheal tube and weaning from invasive ventilation, and thus ends with extubation.

### 2.5. Endpoint

The primary outcome was the incidence of AKI within 7 days after ICU admission, its staging, and the proportions of transient AKI, persistent AKI and AKD.

The secondary endpoints were the determination of the risk factors of transient and persistent AKI, and AKD, and the renal prognostic evaluation assessed with MAKE at the end of ICU and hospital stays and 90 days after ICU admission.

### 2.6. Statistical Analysis

Patient characteristics were expressed as number (percentage) for categorical variables and median (interquartile range (IQR)) for continuous variables. Comparisons were made with exact Fisher tests for categorical variables and Wilcoxon tests for continuous variables.

Univariate followed by multivariate logistic regression analyses were performed to identify risk factors of EO-AKI. We used conditional stepwise variable selection with 0.2 as the critical *p*-value for entry into the model and 0.05 as the *p*-value for removal. Interactions and correlations between the explanatory variables were carefully checked. Longitudinal data were categorized according to their median. Data are given as odds ratio (ORs) and their 95% confidence intervals (CI).

A competing risk analysis was performed to assess cumulative renal recovery. The concomitant competing risks taken into account were discharge alive from the ICU and ICU mortality. The Fine and Gray model [33] was used to assess risk factors of renal recovery. We also used a conditional stepwise variable selection with 0.2 as the critical *p*-value for entry into the model and 0.05 as the *p*-value for removal. It was pre-planned to force the “AKI stage” in the model if this variable was not selected as sensitivity analysis. Data are given as subdistribution hazard ratios (sHRs) and their 95% CI. Proportionality of the risks was also carefully checked.

All reported probability values are two-tailed, and *p*-value < 0.05 was considered to be statistically significant. Missing data were imputed linearly. All analyses were performed with SAS software version 9.4 (SAS Institute Inc., Cary, NC, USA) or R Statistical Software (v4.2.1; R Core Team 2021, Vienna, Austria).

## 3. Results

### 3.1. Initial Characteristics

Of the 336 patients screened, 266 were finally included in our study (Figure 1). The median time between symptom onset and ICU admission was 9 days (IQR, 6–11 d) and between hospital and ICU admission was 2 days (1–5). The median age of the patients was 69 years (61–75), of whom 186 (70%) were of male gender, 50 (18.8%) had a cardiovascular disease and 40 (15%) had diabetes mellitus. Initial severity according to simplified acute physiology score II (SAPS II) and sequential organ failure assessment (SOFA) score was 36 (29–45) and 4 (3–6), respectively; PaO2/FiO2 was 98 (70–210), and 31 (11.6%) patients required vasopressors and invasive mechanical ventilation. Patient characteristics are detailed in Table 1.

### 3.2. Prevalence, Illness Severity Associated with, and Risk Factors for, EO-AKI

The cumulative incidence of EO-AKI and that of EO-AKI recovery, taking into account competing events, are shown in Figure 2, panels A and B, respectively.

On ICU admission, EO-AKI was recorded in 84 patients (31.5%), of whom 42 (50%) had AKI stage 1, 17 (20.2%) had AKI stage 2 and 25 (29.7%) had AKI stage 3 (Figure 1). There was no significant difference in clinical characteristics between patients with and without EO-AKI, except for age (73.7 years (65.2; 76.9) vs. 67.2 years (59.8–73.3), *p* < 0.01), and time from first symptoms to ICU admission (10 days (8–12) vs. 8 days (4.5–10), *p* < 0.01).

On ICU admission, patients with EO-AKI had greater illness severity than patients without as evidenced by a higher SAPS II score (42 (33–50) vs. 34 (28–41), *p* < 0.01) and a greater need for vasopressors (25 (29.8%) vs. 6 (3.3%), *p* < 0.01) and invasive mechanical ventilation (22 (26.2%) vs. 8 (4.4%), *p* < 0.01). They had also received aminoglycosides more often before ICU admission (8 (9.5%) vs. 1 (0.6%), *p* < 0.01) and had higher SCr values (119 µmol/L (86–161) vs. 74 (61–89), *p* < 0.01) and higher serum levels of procalcitonin (0.5 g/L (0.2–2.2) vs. 0.2 g/L (0.1–0.4)).

Other biological inflammatory factors such as serum values of D-dimers, ferritin and C-reactive protein (CRP) were not associated with EO-AKI. The median length of ICU stay was significantly longer in patients with EO-AKI (10.5 days (7–16)) than without (6 days (4–10), *p* < 0.01) (Table 1).

Overall, EO-AKI patients developed greater illness severity than patients without, as shown by worse oxygenation and higher needs in invasive mechanical ventilation and vasopressors (all *p* < 0.01). Higher ICU and hospital mortality rates were observed in patients with EO-AKI (47 (56%) and 52 (61.9%)) than in patients without EO-AKI (32 (17.6%) and 38 (20.8%)), (all *p* < 0.01). The higher the EO-AKI stage, the greater the mortality (Table 2).

Of note, patients with EO-AKI who required RRT during ICU stay, compared to patients who required RRT during ICU but without EO-AKI (late-AKI) were quite similar in terms of age and comorbidities. However, patients with EO-AKI and RRT had more severe disease on ICU admission. The day-90 death rate in these two subgroups was high: 21 (87.5%) for EO-AKI and 9 (81.82%) for late-AKI (Appendix A).

Multivariable analysis showed that risk factors associated with EO-AKI included age, impaired immune system, being under vasopressors on admission, higher serum procalcitonin levels and exposure to aminoglycosides prior to ICU admission. In contrast, we identified that time between symptom onset and ICU admission >10 days was a protective factor against EO-AKI (Figure 3 and Appendix A, Appendix A).

### 3.3. EO-AKI Renal Recovery and Risk Factors

EO-AKI was classified as transient or persistent and AKD in 40 (47.6%), 15 (17.8%) and 29 (34.5%) patients, respectively (Figure 1). The cumulative incidence of EO-AKI recovery in patients with EO-AKI was 54.3% (53.7–54.9) at day 5, and 72.8% (72.3–73.3) at day 20.

Patients who developed AKD had higher ICU and hospital mortality than patients without AKI, 21 (72.4%) vs. 32 (17.6%), (*p* < 0.01) and 23 (79.3%) vs. 38 (20.9%), (*p* < 0.01), respectively. ICU mortality was higher in patients with AKD than in patients with transient and persistent AKI, 21 (72.4%) vs. 7 (46.7%) and 19 (47.5%), respectively, (*p* < 0.01). Patients who developed AKD more often had EO-AKI stage 3 (Table 3).

EO-AKI recovery was independently associated with the initial staging of EO-AKI (stage 3 vs. stage 1, sHR, 0.11; 95% CI, 0.05–0.22), and exposure to diuretics before ICU admission (sHR, 0.39; 95% CI, 0.26–0.6) (Figure 4 and Appendix A).

### 3.4. Outcome of EO-AKI and of EO-AKI Recovery

Patients with EO-AKI had lower survival at day 90 than patients without, 49/76 (64.5%) vs. 38/168 (22.6%) (*p* < 0.01), respectively. The 90-day mortality increased with EO-AKI staging: 22/39 (56.4%), 9/15 (60%) and 18/22 (81.8%) in patients with EO-AKI stages 1, 2 or 3, respectively (*p* < 0.01) (Table 2).

During ICU stay, we identified 38 (14.2%) patients who needed renal replacement therapy, of whom 25 were EO-AKI patients. At day 90, no EO-AKI patients required RRT but 10.8% had a GFR < 60 mL/min.

Among EO-AKI patients, lack of renal recovery was more frequently observed in those with EO-AKI stage 3 (Appendix A). The 90-day mortality was lower in patients with transient or persistent EO-AKI than in patients who developed AKD, 20/36 (55.6%), 8/14 (57.1%) and 21/26 (80.8%) (*p* < 0.01), respectively (Table 3). MAKE-90 occurred in 104/244 (42.6%) of all patients.

## 4. Discussion

To our knowledge, this study is one of the first to characterize EO-AKI in critically ill COVID-19 patients and to assess the relationship between time to recovery and outcome.

First, we confirmed that EO-AKI is a frequent complication of COVID-19 in critically ill patients [15]. The timing of studies according to the outbreak wave and WHO treatment recommendations, the diversity of the case-mix populations (restricted or not to patients on invasive oxygen therapy, for instance), the differences in the definitions of AKI (based on both diuresis and SCr criteria or on SCr criteria alone) and of AKI onset make comparisons between studies on this topic difficult.

In our study, we observed a −31% prevalence of AKI, which is lower than the rates reported in most recent European studies, such as those of Lumlertgul et al. in the United Kingdom [10], who found an AKI prevalence of 76.6%, or of Arrestier et al. [14] in France with an AKI prevalence of 64.3%. In the study by Geri et al. [34], half of the cohort of 379 patients developed AKI in the first 7 days of admission. These studies used a longer time window of up to 21 days, which could be an explanation for the differences in AKI rates. In contrast, in a cohort study performed by the COVID-19 Critical Care Consortium and using the maximum SCr on day 1–2 of invasive mechanical ventilation to define EO-AKI, the rate of EO-AKI was 20.9%.

It is noteworthy that most patients in the SarSCoV-AKI cohort were recruited after August 2020, hence after the first wave in France. As a consequence, they received the recommended treatment of steroids and reinforced anticoagulation, and in most cases the intubation strategy was performed later than during the first wave [35,36]. The patients in our cohort were also probably less severely ill at ICU admission than in cohorts reporting higher rates of AKI with a lesser need for vasopressors, 11.4% compared to 44.3% in Arrestier et al.’s study [15] or 44.2% in Lumlertgul et al.’s study [11]. We also used only SCr results whereas other studies used both SCr and urine output criteria to define AKI. Collectively, the results might have led to an underestimation of AKI prevalence in our study [37].

Older age, a higher procalcitonin level, a higher SOFA score and use of mechanical ventilation were independently associated with an increased risk of EO-AKI in our cohort. These results are consistent with those of other studies in both SARS-CoV-2 ICU populations [10,15,17] and general ICU patients [38,39], in which a higher severity score, hypertension and older age were frequent risk factors for developing AKI.

Since patients who developed AKI had more antimicrobial therapy, especially aminoglycoside use, more organ failure requiring vasopressors compared to patients without AKI, but also presented more bacteriemia on admission, EO-AKI might be caused by sepsis rather than direct renal effects of the SARS-CoV-2 virus in most cases.

Severe COVID-19 is classically associated with elevated values of systemic non-specific inflammatory biomarkers including CRP, ferritin and D-dimers [40,41]. Numerous studies suggest that in critically ill patients, systemic inflammation contributes to AKI occurrence [10,13,42,43]. In our study, only increased values of serum D-dimers were associated with EO-AKI in univariate analysis, but they were no longer found in multivariate models. This absence of an independent association between systemic inflammation and EO-AKI could result from the widespread use of corticosteroids during severe COVID [44]. Further investigations regarding the relationship between the inflammatory process and EO-AKI in severe COVID are therefore warranted.

Although our mortality rate was high at 90 days in EO-AKI patients (64.5%), only 10.8% of them developed chronic kidney disease, as defined by eGFR < 60 mL/min, and none were dialysis dependent. Other studies showed renal recovery rates between 17% and 84% and dialysis dependence rates between 8% and 56.5% [9,45,46,47]. Lumlertgul et al. [10] reported kidney recovery in 90.9% of survivors at 90 days. The absence of dialysis dependence observed in our study could have been due to the exclusion of patients with pre-existing kidney disease. However, these findings strongly suggest that developing AKD after EO-AKI is often a reversible condition in ICU survivors. In our cohort, most of the patients with AKI stages 2 or 3 and persistent AKI or AKD died, and hence, only less severely ill patients survived and were able to recover from EO-AKI.

In our study, which was performed mainly during waves 2 and 3, we observed a lower rate of transient EO-AKI (47.5%) and a higher rate of patients with non-renal recovery ≥7 days (34.5%) than those reported by Lumlertgul et al. during wave 2 (65% and 10%, respectively) [47]. However, patients in the latter study were more likely to have received mechanical ventilation. We observed a higher rate of patients with non-renal recovery ≥7 days in our cohort. We cannot exclude that these discrepancies were partly due to differences in the definition of renal recovery: we used normalization of SCr while Lumlertgul et al. used SCr < 1.5 times baseline value, a more sensitive definition [48]. Additionally, our study shows that patients who made a rapid renal recovery had a better prognosis since AKD was associated with increased mortality at 90 days and mortality was strongly associated with initial AKI stage. These findings are in line with studies in both the general ICU population [49] and COVID-19 ICU populations [15,50].

We also identified that use of diuretics on ICU admission was a risk factor for non-renal recovery. While diuretics can help prevent lung overload, they could have induced hypovolemia and pre-renal AKI, which might increase the risk of renal failure and delay renal adaptation to injury due to SARS-CoV2, and subsequent recovery. On the other hand, we found that an interval between symptom onset and ICU admission greater than 10 days protected against AKI development. This can be explained by the fact that peak infection usually occurs after 3–7 days [51] and ICU admission often occurs before 10 days of the first symptoms [36], and hence, patients admitted to the ICU after 10 days had probably gone through the most unstable period of the disease.

We found that, after adjusting for cofounders, all EO-AKI stages were associated with higher ICU mortality, which is in line with other reports [10,18,52].

Our study shows that EO-AKI patients had a longer length of stay, a higher rate of invasive mechanical ventilation and a greater need for vasopressors. EO-AKI was strongly associated with mortality, which is consistent with the results of other studies evaluating AKI in ICU-hospitalized patients [9,10,14,15,17].

Our study has several strengths. First, we admitted patients consecutively thus limiting selection bias, and had high quality data, prospectively collected, with almost none missing. In addition, we were able to establish the incidence and outcomes of transient AKI, persistent AKI and AKD. To our knowledge, data reporting renal recovery in patients with SARS-CoV2 pneumonia are scarce [10].

However, our work also has certain limitations. First, it was a single-centre observational study, which could impede the generalization of the results. Second, we might have over- or underestimated the prevalence of AKI. True baseline SCr was known for only 55% of patients and if the baseline value was unknown, SCr was estimated using the MDRD formula. In accordance with French legislation, we did not record patient ethnicity, and back-calculated SCr on the assumption that all patients were Caucasians. In addition, we did not define AKI according to the urine output criteria of the KDIGO definition, which could also have biased the true prevalence of AKI. Third, we might have either overestimated the prevalence of AKD, since it was defined by the absence of SCr normalization following EO-AKI and not by a decrease in SCr < 1.5-fold baseline value, or underestimated it since we did not take into account the markers of kidney damage to diagnose AKD. Similarly, since markers of kidney damage were not recorded, the MAKE criterion “persistence of renal failure” was defined by the functional criterion for kidney diseases and disorders: eGFR < 60 mL/min/1.73 m^2^ [28]. Finally, as previously mentioned, follow-up was only 3 months, which is a short period to evaluate renal prognosis, and further studies are needed to evaluate long-term kidney outcomes in COVID ICU survivors after EO-AKI [47,53]. Additional studies to assess long-term renal prognosis should also be undertaken.

## 5. Conclusions

To conclude, we found that one third of critically ill patients with severe COVID-19 developed EO-AKI. EO-AKI stage 3 or AKD were associated with fatal outcome in around two-thirds of the cases at 3 months. EO-AKI stage 1 or transient AKI were also associated with a higher death rate in half of the cases. Monitoring and preserving kidney function should be considered for all patients admitted to the ICU for SARS-CoV-2 pneumonia.

## Figures and Tables

**Figure 1 biomedicines-11-01001-f001:**
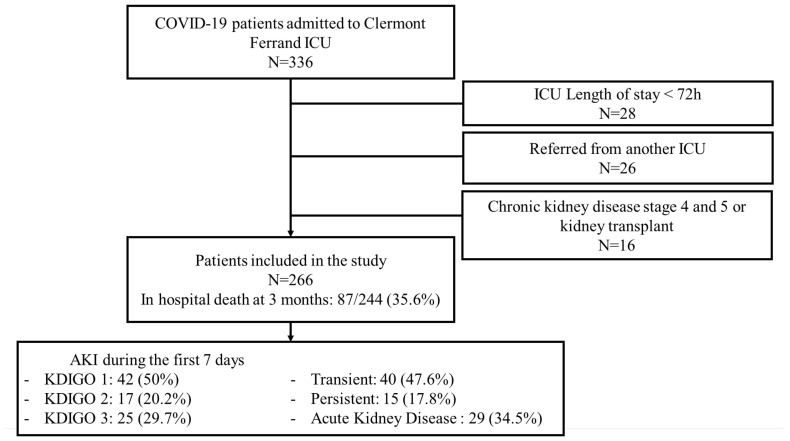
COVID-19 patient flow diagram. ICU: intensive care unit; AKI: acute kidney injury; KDIGO: Kidney Disease Improving Global Outcomes.

**Figure 2 biomedicines-11-01001-f002:**
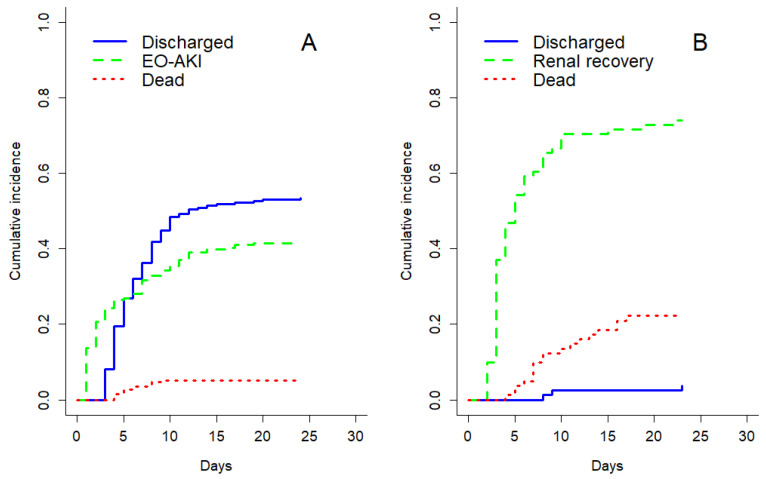
Cumulative incidence of early-onset acute kidney injury (EO-AKI) (**A**) and renal recovery (**B**) taking into account competing risks of discharge alive and mortality.

**Figure 3 biomedicines-11-01001-f003:**
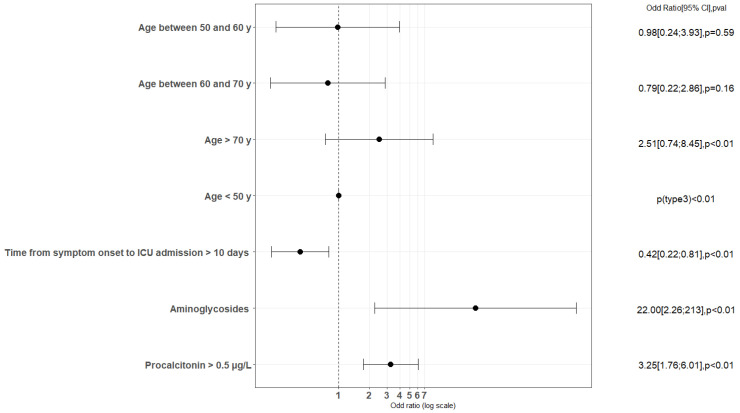
Factors associated with occurrence of AKI, multivariate analyses, logistic regression model. Variables with a *p*-value < 0.1 in the univariate analyses were age, cardiovascular disease, impaired immune system, time from first symptoms to ICU admission, antimicrobial therapy, aminoglycosides, vasopressors, invasive mechanical ventilation, bacteriemia on admission and procalcitonin. Because of correlations, only the following covariates were tested in multivariate analysis: age, cardiovascular disease, impaired immune system, time from first symptoms to ICU admission, aminoglycosides, vasopressors and procalcitonin.

**Figure 4 biomedicines-11-01001-f004:**
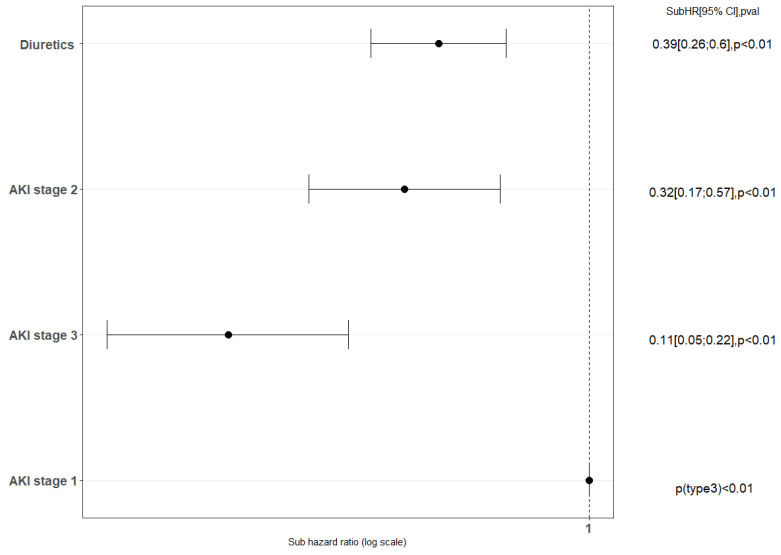
Factors associated with recovery from AKI, multivariate analyses, sub-distribution model. AKI: acute kidney injury; SubHR: sub hazard ratio. Variables with a *p*-value <0.1 in univariate analyses were age, lopinavir–ritonavir, diuretics, vasopressors and KDIGO stage. These covariates were considered into the final model.

**Table 1 biomedicines-11-01001-t001:** Characteristics of critically ill COVID-19 patients according to the occurrence of early-onset acute kidney injury.

VARIABLES, *n* (%) or Median (IQR)	All	No EO-AKI	EO-AKI	*p*-Value
Number of patients	266	182	84	
Period of admission				
Wave 1 (before August 2020)	15 (5.6)	8 (4.4)	7 (8.3)	0.42
Wave 2 (between August 2020 and December 2020)	101 (38)	69 (37.9)	32 (38.1)	
Wave 3 (after January 2021)	150 (56.4)	105 (57.7)	45 (53.6)	
Age, yrs.	68.8 (60.6–74.8)	67.2 (59.8–73.3)	73.7 (65.2–76.9)	<0.01
Sex, men	186 (70)	128 (70.3)	58 (69)	0.83
Comorbidities				
BMI > 30 kg/m^2^	117 (44)	76 (41.8)	41 (48.8)	0.28
Cardiovascular disease	50 (18.8)	29 (15.9)	21 (25)	0.08
Chronic respiratory disease	17 (6.4)	14 (7.7)	3 (3.6)	0.20
Impaired immune system ^1^	43 (16.2)	24 (13.2)	19 (22.6)	0.05
Diabetes mellitus	40 (15)	23 (12.6)	17 (20.2)	0.11
Time from first symptoms to ICU admission (days)	9 (6–11)	10 (8–12)	8 (4.5–10)	<0.01
Time from hospital to ICU admission (days)	2 (1–5)	2 (1–5)	2 (1–5)	0.46
Treatments before ICU				
ACE inhibitors and angiotensin receptor blockers	4 (1.1)	1 (0.5)	3 (3.6)	0.06
Lopinavir–ritonavir	8 (3)	4 (2.2)	4 (4.8)	0.27
Remdesivir	45 (17.2)	36 (20.1)	9 (10.7)	0.06
Steroids	226 (85)	158 (86.8)	68 (81)	0.21
Tocilizumab	25 (9.4)	20 (11)	5 (6)	0.19
Antimicrobial therapy	93 (35.4)	53 (29.6)	40 (47.6)	<0.01
Aminoglycosides	9 (3.4)	1 (0.6)	8 (9.5)	<0.01
Vancomycin	3 (1.2)	2 (1.1)	1 (1.2)	0.96
Diuretics	207 (77.8)	142 (78)	65 (77.4)	0.91
Organ failures at ICU admission				
SAPS II	36 (29–45)	34.5 (28–41)	42 (33–50)	<0.01
SOFA	4 (3–6)	4 (3–5)	5 (4–9)	<0.01
PaO2/FiO2	98 (70–210)	98 (71–248)	90 (67–190)	0.20
Need for vasopressors	31 (11.6)	6 (3.3)	25 (29.8)	<0.01
Invasive mechanical ventilation	30 (11.2)	8 (4.4)	22 (26.2)	<0.01
Renal replacement therapy	3 (1.2)	0 (0)	3 (3.6)	0.01
Pulmonary bacterial co-infection	18 (6.8)	12 (6.7)	6 (7.1)	0.90
Bacteriemia on admission	11 (4.2)	4(2.2)	7(8.3)	0.02
Admission laboratory profile				
Neutrophils (NA = 20), G/L	6.6 (4.8–9.8)	6.4 (4.8–8.9)	7.6 (4.5–11.9)	0.04
Lymphocytes (NA = 19), G/L	0.6 (0.4–1)	0.7 (0.5–1)	0.6 (0.4–0.9)	0.50
Creatininemia, µmol/L	81.6 (65–105)	73.5 (61–89)	119 (85.5–161)	<0.01
Procalcitonin (NA = 17), g/L	0.2 (0.2–0.6)	0.2 (0.1–0.4)	0.5 (0.2–2.2)	<0.01
C Reactive protein (NA = 52), mg/L	119 (70–170)	113 (70–163)	125 (82–171)	0.73
D-dimers (NA = 6), µg/L	1214 (777–2027)	1168 (727–1939)	1216 (1000–2900)	0.05
Fibrinogen (NA = 4), g/L	7.2 (6.2–8)	7.2 (6.5–8)	7 (5.8–7.9)	0.18
Ferritin (NA = 19), ng/mL	1144 (644–1945)	1130 (684–1850)	1177 (584–2142)	0.58
Organ failures during ICU stay				
Need for vasopressors	85 (32)	33 (18.1)	52 (61.9)	<0.01
Invasive mechanical ventilation	88 (33)	35 (19.2)	53 (63.1)	<0.01
Renal replacement therapy	38 (14.2)	13 (7.1)	25 (29.8)	<0.01
General outcomes				
Pulmonary embolism	14 (5.2)	9 (5)	5 (6)	0.74
Ventilator-associated pneumonia	35 (13.4)	13 (7.3)	22 (26.2)	<0.01
Decision not to intubate during ICU stay	40 (15)	18 (9.9)	22 (26.2)	<0.01
Length of ICU stay, d	8 (5–13)	6 (4–10)	10.5 (7–16)	<0.01
Death in ICU	79 (29.6)	32 (17.6)	47 (56)	<0.01
Length of hospital stay, d	14 (10–23)	14 (10–24)	15 (9.5–22.5)	0.50
In-hospital death	90 (33.8)	38 (20.8)	52 (61.9)	<0.01
RRT at hospital discharge	1 (0.6)	1 (0.8)	0 (0)	0.63
General outcomes at 90 days (MAKE-90)				
eGFR < 60 mL/min at 90 days	17/158 (10.8)	9/131 (6.9)	8/27 (29.6)	<0.01
RRT at 90 days	0/163	0/132	0/31	
Death at 90 days	87/244 (35.6)	38/168 (22.6)	49/76 (64.5)	<0.01
MAKE-90	104/244(42.6)	47/168 (28)	57/76 (75)	<0.01

ACE: angiotensin converting enzyme—AKI, acute kidney injury; BMI, body mass index; ECMO, extracorporeal membrane oxygenation; GFR, glomerular filtration rate; ICU, intensive care unit; NSAID, non-steroidal anti-inflammatory drug; RRT, renal replacement therapy; SAPS II, simplified acute physiology score; SOFA, sequential organ failure assessment. ^1^ Aplasia (lymphocytes < 1 G/L; or corticosteroids (if treatment duration > 1 month or if treatment amount > 2 mg/kg regardless of duration); or HIV (positive serology); AIDS (positive HIV serology and clinical complications: pneumocystis pneumonia, Kaposi’s sarcoma, tuberculosis, toxoplasmosis.

**Table 2 biomedicines-11-01001-t002:** Characteristics of the critically ill COVID-19 patients according to the severity of the acute kidney injury (KDIGO stage).

VARIABLES, *n* (%) or Median (IQR)	No AKI	AKI Stage 1	AKI Stage 2	AKI Stage 3	*p*-Value
Number of patients	182	42	17	25	
Period of admission					
Wave 1 (before August 2020)	8 (4.4)	2 (4.8)	1 (5.9)	4 (16)	0.37
Wave 2 (between August 2020 and December 2020)	69 (37.9)	18 (42.9)	7 (41.2)	7 (28)	
Wave 3 (after January 2021)	105 (57.7)	22 (52.4)	9 (52.9)	14 (56)	
Age, yrs.	67.2 (59.8–73.3)	74.5 (66.4–78.2)	74.7 (64.7–77.4)	72.7 (63.4–75.1)	<0.01
Sex, men	128 (70.3)	30 (71.4)	8 (47.1)	20 (80)	0.14
BMI, kg/m^2^	28.7 (25.1–32.1)	27.4 (24.9–31.1)	28.3 (26.9–34.7)	30.8 (25.5–36)	0.52
Comorbidities, no (%)					
BMI > 30 kg/m^2^	76 (41.8)	18 (42.9)	8 (47.1)	15 (60)	0.38
Cardiovascular disease	29 (15.9)	9 (21.4)	4 (23.5)	8 (32)	0.23
Chronic respiratory disease	14 (7.7)	1 (2.4)	2 (11.8)	0	0.24
Chronic kidney disease	2 (1.1)	5 (11.9)	0	3 (12)	<0.01
Impaired immune system ^1^	24 (13.2)	10 (23.8)	2 (11.8)	7 (28)	0.12
Diabetes mellitus	23 (12.6)	9 (21.4)	1 (5.9)	7 (28)	0.09
Time from first symptoms to ICU admission	10 (8–12)	8.5 (4–11)	9 (5–11)	6 (5–10)	<0.01
Treatment before ICU					
ACE inhibitors and angiotensin receptor blockers	1 (0.6)	2 (4.8)	1 (5.9)	0	0.08
Lopinavir–ritonavir	4 (2.2)	1 (2.4)	1 (5.9)	2 (8)	0.39
Remdesivir	36 (20.1)	6 (14.3)	1 (5.9)	2 (8)	0.23
Steroids	158 (86.8)	38 (90.5)	14 (82.4)	16 (64)	0.02
Tocilizumab	20 (11)	2 (4.8)	1 (5.9)	2 (8)	0.59
Antimicrobial therapy	53 (29.6)	20 (47.6)	7 (41.2)	13 (52)	0.04
Aminoglycosides	1 (0.6)	3 (7.1)	1 (5.9)	4 (16)	<0.01
Vancomycin	2 (1.1)	0	0	1 (4)	0.48
Diuretics	142 (78)	36 (85.7)	11 (64.7)	18 (72)	0.30
Organ failures at ICU admission					
SAPS II	34.5 (28–4)	40.5 (3–47)	42 (31–50)	47 (34–56)	<0.01
SOFA	4 (3–5)	5 (4–7)	7 (5–8)	9 (4–12)	<0.01
PaO2/FiO2	98 (71–247)	82 (67–176)	123 (65–243)	98 (67–141)	0.50
Need for vasopressors	6 (3.3)	6 (14.3)	6 (35.3)	13 (52)	<0.01
Invasive mechanical ventilation	8 (4.4)	6 (14.3)	6 (35.3)	10 (40)	<0.01
Renal replacement therapy	0	0	0	3 (12)	<0.01
Pulmonary bacterial co-infection	12 (6.7)	1 (2.4)	0	5 (20)	0.03
Bacteriemia on admission	4(2.23)	4(9.52)	0	3(12)	0.03
Admission laboratory profile					
Neutrophils (NA = 20), G/L	6.4 (4.8–8.9)	7.5 (5.5–11.9)	5.5 (3.7–14.4)	7.8 (4.1–11.8)	0.19
Lymphocytes (NA = 19), G/L	0.7 (0.49–1.04)	0.57 (0.4–0.77)	0.86 (0.69–1.64)	0.57 (0.36–0.9)	0.01
Creatininemia, µmol/L	74 (61–89)	121 (97–135)	91 (64–156)	119 (85–202)	<0.01
Procalcitonin (NA = 17), g/L	0.18 (0.11–0.44)	0.38 (0.23–3.69)	0.31 (0.15–0.99)	0.71 (0.26–2.24)	<0.01
C Reactive protein (NA = 52), mg/L	113 (70–163)	121 (83–158)	108 (53–147)	150 (45–214)	0.72
D-dimers (NA = 6), µg/L	1168 (727–1939)	1198 (953–2531)	1926 (1143–3381)	1308 (927–3730)	0.19
Fibrinogen (NA = 4), g/L	7.2 (6.5–8)	7.2 (6.4–7.9)	6.3 (5.3–7.5)	7 (6–7.9)	0.27
Ferritin (NA = 19), ng/mL	1130 (684–1850)	1129 (528–1846)	1253 (535–2429)	1258 (681–2753)	0.92
Organ failures during ICU stay					
Need for vasopressors	33 (18.1)	18 (42.9)	13 (76.5)	21 (84)	<0.01
Invasive mechanical ventilation	35 (19.2)	18 (42.9)	14 (82.4)	21 (84)	<0.01
Pulmonary embolism	9 (5.0)	3 (7.1)	1 (5.9)	1 (4)	0.94
Ventilator-associated pneumonia	13 (7.3)	9 (21.4)	6 (35.3)	7 (28)	<0.01
Type of Renal Recovery, no (%)					
Transient		32 (76.2)	6 (35.3)	2 (8)	<0.01
Persistent		7 (16.7)	6 (35.3)	2 (8)	
Acute kidney disease		3 (7.1)	5 (29.4)	21 (84)	
General Outcomes					
Renal replacement therapy	13 (7.1)	6 (14.3)	4 (23.5)	15 (60)	<0.01
Decision not to intubate during ICU stay	18 (9.9)	12 (28.6)	5 (29.4)	5 (20)	0.01
Length of ICU stay, d	6 (4–10)	9.5 (6–15)	11 (7–16)	13 (8–19)	<0.01
Death in ICU	32 (17.6)	19 (45.2)	9 (52.9)	19 (76)	<0.01
Length of hospital stay, d	14 (10–24)	15.5 (10–22)	16 (9–35)	15 (10–21)	0.86
In-hospital death	38 (20.9)	22 (52.4)	9 (52.9)	21 (84)	<0.01
RRT at hospital discharge	1 (0.75)	0	0	0	0.97
General Outcomes at 90 days (MAKE-90)					
eGFR < 60 mL/min at 90 days	9/131	6/17	1/6	1/4	<0.01
RRT at 90 days	0/132	0/19	0/7	0/5	
Death at 90 days	38/168 (22.6)	22/39 (56.4)	9/15 (60)	18/22 (81.8)	<0.01
MAKE-90	47/168 (28.0)	28/39 (71.8)	10/15 (66.7)	19/22 (86.4)	<0.01

ACE, angiotensin converting enzyme; AKI, acute kidney injury; BMI, body mass index; ECMO, extra corporeal membrane oxygenation; GFR, glomerular filtration rate; ICU, intensive care unit; NSAID, non-steroidal anti-inflammatory drug; RRT, renal replacement therapy; SAPS II, simplified acute physiology score; SOFA, sequential organ failure assessment. ^1^ Aplasia (lymphocytes < 1 G/L); or corticosteroids (if treatment duration > 1 month or if treatment amount > 2 mg/kg regardless of duration); or HIV (positive serology); AIDS (positive HIV serology and clinical complications: pneumocystis pneumonia, Kaposi’s sarcoma, tuberculosis, toxoplasmosis.

**Table 3 biomedicines-11-01001-t003:** Characteristics of the critically ill COVID-19 patients according to renal recovery.

VARIABLES, *n* (%) or Median (IQR)	No AKI	Transient AKI	Persistent AKI	Acute Kidney Disease	*p*-Value
Number of patients	182	40	15	29	
Period of admission					
Wave 1 (before August 2020)	8 (4.4)	1 (2.5)	1 (6.7)	5 (17.2)	0.10
Wave 2 (between August 2020 and December 2020)	69 (37.9)	14 (35)	8 (53.3)	10 (34.5)	
Wave 3 (after January 2021)	105 (57.7)	25 (62.5)	6 (40)	14 (48.3)	
Age, yrs.	67.2 (59.8–73.3)	73.9 (65.2–77.8)	73.8 (68.8–75.9)	72.81 (63.4–76.4)	<0.01
Sex, men	128 (70.3)	27 (67.5)	9 (60)	22 (75.9)	0.73
BMI, kg/m^2^	28.7 (25.1–32.1)	27.44 (24.0–31.6)	28.3 (26.3–32)	30.8 (26.1–36.0)	0.57
Comorbidities					
BMI > 30 kg/m^2^	76 (41.8)	17 (42.5)	7 (46.7)	17 (58.6)	0.40
Cardiovascular disease	29 (15.9)	10 (25)	6 (40)	5 (17.2)	0.09
Chronic respiratory disease	14 (7.7)	2 (5)	0	1 (3.5)	0.55
Chronic kidney disease	2 (1.1)	6 (15)	0	2 (6.9)	<0.01
Impaired immune system ^1^	24 (13.2)	8 (20)	4 (26.7)	7 (24.1)	0.24
Diabetes mellitus	23 (12.6)	7 (17.5)	2 (13.3)	8 (27.6)	0.20
Time from first symptoms to ICU admission, d	10 (8–12)	8 (4–10.5)	9 (4–11)	7 (5–10)	<0.01
Treatment before ICU					
ACE inhibitors and angiotensin receptor blockers	1 (0.6)	1 (2.5)	1 (6.7)	1 (3.5)	0.19
Lopinavir–ritonavir	4 (2.2)	0	1 (6.7)	3 (10.3)	0.06
Remdesivir	36 (20.1)	6 (15)	1 (6.7)	2 (6.9)	0.21
Steroids	158 (86.8)	36 (90)	11 (73.3)	21 (72.4)	0.09
Tocilizumab	20 (11)	4 (10)	0	1 (3.5)	0.35
Antimicrobial therapy	53 (29.6)	19 (47.5)	7 (46.7)	14 (48.3)	0.04
Aminoglycosides	1 (0.6)	3 (7.5)	0	5 (17.2)	<0.01
Vancomycin	2 (1.1)	0	0	1 (3.4)	0.57
Diuretics	142 (78)	30 (75)	12 (80)	23 (79.3)	0.97
Organ failures at ICU admission					
SAPS II	34.5 (28–41)	40.5 (31–46.5)	47 (38–58)	44 (34–54)	<0.01
SOFA	4 (3–5)	5 (4–7)	6 (5–10)	8 (4–10)	<0.01
PaO2/FiO2	98 (71.3–247.6)	82.5 (70–220)	104 (66–200)	86.7 (63.8–138.8)	0.59
Need for vasopressors	6 (3.3)	8 (20)	5 (33.3)	12 (41.4)	<0.01
Invasive mechanical ventilation	8 (4.4)	9 (22.5)	2 (13.3)	11 (37.9)	<0.01
ECMO	0	0	0	1 (3.45)	0.04
Pulmonary bacterial co-infection	12 (6.7)	0	2 (13.3)	4 (13.8)	0.11
Bacteriemia on admission	4 (2.23)	4(10)	0	3(10.34)	0.04
Admission laboratory profile					
Neutrophils (NA = 20), G/L	6.37 (4.76–8.94)	7.24 (4.63–11.2)	9.75 (5.1–14.44)	7.59 (4.09–13.32)	0.15
Lymphocytes (NA = 19), G/L	0.7 (0.49–1.04)	0.64 (0.44–0.82)	0.77 (0.37–1.33)	0.62 (0.4–0.88)	0.71
Creatininemia, µmol/L	74 (61–89)	122 (92.5–147)	122 (88–190)	105 (81–176)	<0.01
Procalcitonin (NA = 17), g/L	0.18 (0.11–0.44)	0.38 (0.18–3.43)	0.71 (0.31–3.86)	0.56 (0.22–2)	<0.01
C Reactive protein (NA = 52), mg/L	113 (70–163)	117 (82–147)	108 (83–214)	135 (45–175)	0.92
D-dimers (NA = 6), µg/L	1168 (727–1939)	1217 (841.5–2542)	1209 (1000–3961)	1241 (1100–2041)	0.20
Fibrinogen (NA = 4), g/L	7.2 (6.5–8)	7.2 (5.8–7.7)	7.9 (6–9.5)	7 (5.8–7.7)	0.31
Ferritin (NA = 19), ng/mL	1130 (684–1850)	1129 (528–2142)	1191 (538–1831)	1391 (701–2720)	0.72
Organ failures during ICU stay					
Need for vasopressors	33 (18.1)	20 (50)	9 (60)	23 (79.3)	<0.01
Invasive mechanical ventilation	35 (19.2)	21 (52.5)	8 (53.3)	24 (82.8)	<0.01
Pulmonary embolism	9 (5)	3 (7.5)	1 (6.7)	1 (3.5)	0.88
Ventilator-associated pneumonia	13 (7.3)	11 (27.5)	3 (20)	8 (27.6)	<0.01
Severity of Acute Kidney Injury					
KDIGO 1		32 (80)	7 (46.7)	3 (10.3)	<0.01
KDIGO 2		6 (15)	6 (40)	5 (17.2)	
KDIGO 3		2 (5)	2 (13.3)	21 (72.4)	
General outcomes					
Renal replacement therapy	13 (7.1)	5 (12.5)	3 (20)	17 (58.6)	<0.01
Decision not to intubate during ICU Stay	18 (9.9)	12 (30)	3 (20)	7 (24.1)	<0.01
Length of ICU stay, d	6 (4–10)	9 (6–15.5)	10 (8–13)	13 (8–19)	<0.01
Death in ICU	32 (17.6)	19 (47.5)	7 (46.7)	21 (72.4)	<0.01
Length of hospital stay, d	14 (10–24)	15.5 (10.5–24)	16 (9–31)	15 (10–21)	0.87
In-hospital death	38 (20.9)	21 (52.5)	8 (53.3)	23 (79.3)	<0.01
RRT at hospital discharge	1 (0.75)	0	0	0	0.97
General outcomes at 90 days (MAKE-90)					
eGFR < 60 mL/min at 90 days	9/131	6/16	1/6	1/5	<0.01
RRT at 90 days	0/132	0/18	0/6	0/7	
Death at 90 days	38/168 (22.6)	20/36 (55.6)	8/14 (57.1)	21/26 (80.8)	<0.01
MAKE-90	47/168 (28.0)	26/36 (72.2)	9/14 (64.3)	22/26 (84.6)	<0.01

ACE, angiotensin converting enzyme; AKI, acute kidney injury; BMI, body mass index; ECMO, extra corporeal membrane oxygenation; GFR, glomerular filtration rate; ICU, intensive care unit; NSAID, non-steroidal anti-inflammatory drug; RRT, renal replacement therapy; SAPS II, simplified acute physiology score; SOFA, sequential organ failure assessment. ^1^ Aplasia (lymphocytes < 1000/mm^3^); or corticosteroids (if treatment duration > 1 month or if treatment amount > 2 mg/kg regardless of duration); or HIV (positive serology); AIDS (positive HIV serology and clinical complications: pneumocystis pneumonia, Kaposi’s sarcoma, tuberculosis, toxoplasmosis.

## Data Availability

The data can be provided under request at the corresponding author.

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
