# Peer review of "Epidemiology and Outcome of Early-Onset Acute Kidney Injury and Recovery in Critically Ill COVID-19 Patients: A Retrospective Analysis"

_biomedicines, 2023, doi:10.3390/biomedicines11041001_

Round 1

Author Response

  1. Why did the authors use EO-AKI? What are the differences between EO-AKI and AKI?

Thanks for this remark. We understand your point. We preferred to use in this manuscript EO-AKI in order to underline that we only consider the AKI that occurred during the first week after ICU admission, using the last definition of AKI (ref Lameire Kidney international 2021). In the manuscript, the definition of EO-AKI is as followed: “AKI occurring within 7 days after ICU admission: only the first KI episode was taken into account in the study”.

  1. Did the patients admit directly to ICU? Or Did the patients admit to a general ward and then move to ICU because of their condition?

The patients spent in median 2 days [1; 5] in a general ward before ICU admission. We had this information into the Table 1 and into the first part of the results.

  1. Could you assess the ethnicity according to the risk factors of AKI or include the limitation?

In France, it is not allowed to collect the ethnicity of the patients, therefore we didn’t have this information. Furthermore, the following sentence dealing with this remark is into the discussion: “In accordance with French legislation, we did not record ethnicity, and back-calculated Scr on assumption that all patients were Caucasians.”

  1. The authors would add the R version.

Done

  1. In Figures 3 and 4, what are the color differences? à OK, we now provide the figure only in black.
  2. Please make uniform the number of significant digits. à Ok for all the results, we now make uniform the number of significant digits.
  3. Please make sure that the first time the authors use an abbreviation, it's important to spell out the full term and put the abbreviation in parentheses à OK, done, thanks for the remark.

Reviewer 2 Report

Even if at this point COVID-19 pandemic seems to be over, the consequences of SARS-CoV-2 infection still represent a major concern for the health system, aspect confirmed by the results of your study. The whole study is consistently presented, highlighting the need of an adequate renal function assessment in patients diagnosed with SARS-CoV-2 pneumonia, especially in critical ill population. In my opinion, no additional modifications or explanations are required.

Author Response

thank you very much for the nice comment you provided. 

Reviewer 3 Report

Ruault A et al studied the impact of early-onset acute kidney injury (EO-AKI) and recovery in 266 patients with severe COVID-19 infection admitted to ICU. They compared 84 patients with EO-AKI with 182 without EO-AKI. EO-AKI was transient, persistent, and AKD in 40 (47.6%), 15 (17.8%), and 29 (34.6%) patients, respectively. 90-day mortality was 35.6% and increased with EO-AKI occurrence and severity (at stage 3, 81.8% mortality). 90-day mortality in patients with transient or persistent AKI and AKD was 55.6%, 57.1%, and 80.8%, respectively (p<0.01). MAKE-90 occurred in 42.6% of all patients.

They concluded that in ICU patients admitted for SARS-CoV-2 pneumonia, the development of EO-AKI and time to recovery beyond day 7 of onset were associated with poor outcome.

General comment

The Authors faced a poorly debated issue, and the message coming from this narrative review is very interesting. To my mind, the paper is well written.

Major points:

1)      I see from Table 1 that 38 patients underwent RRT, most of them in the group EO-AKI (25, at about 30%). The Author wrote in Discussion lines 293-296: “developing  AKD after EO-AKI is often a reversible condition in ICU survivors. In our cohort, most of  the patients with AKI stages 2 or 3 and persistent AKI or AKD died, and hence only less severely ill patients survived and were able to recover from EO-AKI.” 

As mortality in Covid patients requiring KRP was reported very high (and this is the experience of the Authors too), I’m curious to see if the RKT-treated patients with EO-AKI had different main baseline characteristics and outcome different from those with late onset-AKI. Please provide a new Table for these subgroups of patients.

Minor points

1) In the Abstract the Authors detailed the data as percentages and no absolute figures. Please provide also the total number of studied patients and some figures for categorized patients.

Author Response

Reviewer 3 à Ruault A et al studied the impact of early-onset acute kidney injury (EO-AKI) and recovery in 266 patients with severe COVID-19 infection admitted to ICU. They compared 84 patients with EO-AKI with 182 without EO-AKI. EO-AKI was transient, persistent, and AKD in 40 (47.6%), 15 (17.8%), and 29 (34.6%) patients, respectively. 90-day mortality was 35.6% and increased with EO-AKI occurrence and severity (at stage 3, 81.8% mortality). 90-day mortality in patients with transient or persistent AKI and AKD was 55.6%, 57.1%, and 80.8%, respectively (p<0.01). MAKE-90 occurred in 42.6% of all patients.

They concluded that in ICU patients admitted for SARS-CoV-2 pneumonia, the development of EO-AKI and time to recovery beyond day 7 of onset were associated with poor outcome.

General comment

The Authors faced a poorly debated issue, and the message coming from this narrative review is very interesting. To my mind, the paper is well written.

Major points:

1)      I see from Table 1 that 38 patients underwent RRT, most of them in the group EO-AKI (25, at about 30%). The Author wrote in Discussion lines 293-296: “developing  AKD after EO-AKI is often a reversible condition in ICU survivors. In our cohort, most of  the patients with AKI stages 2 or 3 and persistent AKI or AKD died, and hence only less severely ill patients survived and were able to recover from EO-AKI.” 

As mortality in Covid patients requiring KRP was reported very high (and this is the experience of the Authors too), I’m curious to see if the RKT-treated patients with EO-AKI had different main baseline characteristics and outcome different from those with late onset-AKI. Please provide a new Table for these subgroups of patients.

 We added into the supplementary, a new table reporting the description of the patients without AKI, with EO-AKI and late-AKI and describe among those sub groups the patients who benefitted from RRT or no RRT.

Patients with EO-AKI that needed RRT compared to patients with late-RRT and  RRT, had similar age, and comorbidities, they tended however to be more obese, and most of all were more severe on ICU admission with a higher SAPS II score and SOFA score. All of those patients whatever the timing of their AKI, had a very high death rate with 87.5% and  81.8% of day-90 mortality rate for EO-AKI and Late -AKI respectively.  A sentence was added into the results and the discussion about this new result.

Minor points

1) In the Abstract the Authors detailed the data as percentages and no absolute figures. Please provide also the total number of studied patients and some figures for categorized patients.

Ok, done

Reviewer 4 Report

In this manuscript, authors showed that the development of early-onset acute kidney injury (EO-AKI) as well as longer time to recovery from EO-AKI were significantly associated with poor outcome such as 90 day mortality in critically ill patients with SARS-CoV-2 pneumonia in a single center retrospective study. The subject of study seems to be interesting, and the study was well designed. However, there are some unclear points in this manuscript. The reviewer’s comments are described as follows.

1. Authors described that pneumonia was confirmed by computed tomography in the study design, and that bacterial pneumonia was diagnosed in 6.8 of subjects in Table 1. Thus, this study included acute respiratory failure patients with not only SARS-CoV-2 pneumonia but also typical community-based pneumonia. How authors defined bacterial pneumonia or SARS-CoV-2 pneumonia in this study should be explained.

2. Since patients who developed AKI had more antimicrobial therapy, especially aminoglycoside use, and more organ failure requiring vasopressors compared to patients without AKI, AKI might be caused by sepsis rather than direct renal effects of SARS-CoV-2 virus in most cases. Did AKI patients more frequently have any bacterial infections other than pneumonia?

3. In multivariable analysis in Figure 3, authors should explain why these factors (age, immunosuppression, vasopressor use at admission, higher procalcitonin, and aminoglycoside use) were sufficient for adjustment. Can other factors such as mechanical ventilation be excluded as adjusting factors?

4. In Figure 4, were adjusting factors same as in Figure 3? Authors should specifically describe the adjusting factors.

5. In Tables, does “converting enzyme inhibitors” mean only ACE inhibitors or include both ACE inhibitors and angiotensin receptor blockers?

Author Response

1.Authors described that pneumonia was confirmed by computed tomography in the study design, and that bacterial pneumonia was diagnosed in 6.8 of subjects in Table 1. Thus, this study included acute respiratory failure patients with not only SARS-CoV-2 pneumonia but also typical community-based pneumonia. How authors defined bacterial pneumonia or SARS-CoV-2 pneumonia in this study should be explained. We had into the methodology part of the manuscript the definition we used for bacterial pneumonia on admission.

Pulmonary bacterial co-infection was defined by the presence of a community-acquired or hospital-acquired bacterial pneumonia associated with SARS-CoV2 pneumonia during the ICU admission. The presence of pulmonary bacterial co-infection was defined by the presence of radiological and/or scanographic condensation, bacteriological documentation (a positive quantitative culture of lower respiratory tract samples collected as recommended (bronchoalveolar lavage, >104 CFU/mL, plugged telescoping catheter, >103 CFU/mL, endotracheal aspirate, >106 CFU/mL) and/or presence of positive antigenuria, as defined by the European Centre for Disease Control and Prevention[31]. If bacterial pneumonia occurred at least 2 days after intubation, they were classified ventilator associated pneumoniae (VAP)[32]. The period at risk for VAP begins from 48h after intubation, until removal of the tracheal tube and weaning from the invasive ventilation, so it ends with extubation.

  1. Since patients who developed AKI had more antimicrobial therapy, especially aminoglycoside use, and more organ failure requiring vasopressors compared to patients without AKI, AKI might be caused by sepsis rather than direct renal effects of SARS-CoV-2 virus in most cases. Did AKI patients more frequently have any bacterial infections other than pneumonia?

à We now add the occurrence of bacteriemia recorded during the first two days after ICU admission. Patients with EO-AKI add more bacteriemia on admission. We discuss this new result into the discussion.

“Since patients who developed AKI had more antimicrobial therapy, especially aminoglycoside use, more organ failure requiring vasopressors compared to patients without AKI, but also presented more bacteriemia on admission, EO-AKI might be caused by sepsis rather than direct renal effects of SARS-CoV-2 virus in most cases.”

  1. In multivariable analysis in Figure 3, authors should explain why these factors (age, immunosuppression, vasopressor use at admission, higher procalcitonin, and aminoglycoside use) were sufficient for adjustment. Can other factors such as mechanical ventilation be excluded as adjusting factors?

Only these factors remained statistically significantly associated with occurrence of EO-AKI in our multivariate analysis.

Results of the univariate analyses are now reported into the supplementary data.

  1. In Figure 4, were adjusting factors same as in Figure 3? Authors should specifically describe the adjusting factors.

Thanks for the remark, we now add into the supplementary data, the univariate analyses and the concordance table to explain how we chosen our final model.

  1. In Tables, does “converting enzyme inhibitors” mean only ACE inhibitors or include both ACE inhibitors and angiotensin receptor blockers?

Ok, thanks for the remark, we replace now “converting enzyme inhibitors” by “ACE inhibitors and angiotensin receptor blockers” into the tables

Round 2

Reviewer 3 Report

I have no further comments

Reviewer 4 Report

Authors have successfully addressed all of the reviewer's concerns. There are no more comments.